# Struvite as P Fertilizer on Yield, Nutrient Uptake and Soil Nutrient Status in the Rice–Wheat Rotation System: A Two-Year Field Observation

Jizheng Wang [1,2], Lihong Xue [1,2,*], Pengfu Hou [1,2], Tianjia Hao [1,2], Lixiang Xue [2], Xi Zhang [2], Tianyi Sun [1,2], Sergey Lobanov [3] and Linzhang Yang [2]

1 Nanjing Agricultural University, Nanjing 210095, China; wjz907881854@163.com (J.W.); houpengfu@jaas.ac.cn (P.H.); haotianjia20230313@163.com (T.H.); suntianyi980@gmail.com (T.S.)
2 Institute of Agricultural Resources and Environment, Jiangsu Academy of Agricultural Sciences, Zhongling Street 50, Nanjing 210014, China; njxuelx001@163.com (L.X.); lzyang@issas.ac.cn (L.Y.)
3 Shanghai Phosmag Environmental Technology Co. Ltd., Shanghai 200438, China; lobanov@daad-alumni.de
* Correspondence: njxuelihong@gmail.com

**Abstract:** Long-term large inputs of phosphorus (P) fertilizer in China have caused serious soil P accumulation, low P use efficiency (PUE) and high risk of P loss. Controlling the amount of P fertilizer applied presents an inevitable choice for improving the PUE. Struvite recycled from agricultural wastewater rich in N and P concentrations are capable of slow nutrient release, improving nutrient uptake and enabling the reuse of nutrients from environmental sources when applied to agricultural land. A two-year field experiment was conducted to investigate the effects of struvite combined with P reduction under a rice–wheat rotation system in eastern China. A total of five treatments were set up, including conventional fertilization (FP), a struvite substitution of 100% P fertilizer (SP), a 50% P reduction with struvite substitution (RSP), no application of N fertilizer (N0) and no application of P fertilizer (P0). Grain yield, crop N and P uptake, N and P use efficiency (NUE and PUE) and soil nutrient status were assessed. Under the same P application rate, the yield and aboveground biomass of the SP treatment were slightly higher than those of FP treatment, but the crop P uptake, PUE and soil available P content were significantly increased. The RSP treatment did not reduce yield with 50% P reduction, and significantly improved the PUE and soil available P content. Crop N uptake and NUE were also found to be increased in SP and RSP treatments with struvite substitution. The P apparent balance showed that both the SP and FP treatments had a P surplus, but the RSP treatment had a P break-even, and the soil available P content remains stable compared with the initial value. The results indicate that struvite application could improve the soil P availability and crop nutrient uptake then promote the crop yield. To increase the nutrient use efficiency of crops while ensuring crop yield and soil fertility, appropriate P reduction combined with struvite as a P fertilizer could be sustainable in the rice–wheat rotation system in the long run.

**Keywords:** struvite; P reduction; grain yield; nutrient use efficiency; P apparent balance; rice–wheat rotation

## 1. Introduction

Rice and wheat are the most important cereal crops in the world. Rice–winter wheat double-cropping rotation is a main cultivation mode in the Asian subtropics including the middle and lower reaches of the Yangtze River in China. To meet the food demand of the rapid population, chemical fertilizer has been applied to farmlands worldwide to enhance agricultural productivity, especially in the developed regions in China. The average phosphorus (P) application rate was 100 kg/ha in the wheat season and 75 kg/ha in rice season in the lower reaches of the Yangtze River region, China [1]. However, the nutrient balance analysis showed that the P surplus in wheat were common and accounted for about

33.4–73.9% of the P fertilizer applied, while accounted for 21.6% of the P applied in the rice season in the middle reaches of the Yangtze River [1]. Long-term excessive P fertilizer input under rice–wheat rotation systems in China led to the serious accumulation of soil P, and the available P content of the paddy soils increased significantly at an annual average rate of 11% [2]. In 2019, the average available P content of the paddy soils in Jiangsu was as high as 29.9 mg/kg, and even as high as 60–70 mg/kg in some areas [3]. The high input of P fertilizer resulted in the low P use efficiency (PUE) of only 10–20% [4], also increased the P loss risks through runoff and leaching, which caused the water eutrophication. When soil available P is higher than 40 mg/kg, the risk of P loss will threaten the water environment, especially in the southern of China with a dense regional water network and abundant rainfall [5]. Considering the scarce P resources, how to improve the PUE while ensuring crop yield is urgent for sustainable agricultural production and non-point pollution control.

Nutrient management optimizing, as one of the most effective ways for non-point pollution control, has become a hotspot of research worldwide. However, studies to date have mainly focused on N rate reduction strategies for maintaining yield and reducing N losses [6]. Little attention was given to P rate reduction. Song et al. (2019) [7] found that P reduction combined with dry and wet alternating irrigation improved PUE and reduced the P loss without causing rice yield loss. Wang et al. (2016) [8] reported that only P fertilization in wheat season is feasible to maintain the yield, improve the PUE and reduce the P loss risk in the rice–wheat rotation system.

Except for the adjustment of P rate according to the soil P status and crop needs, slow/controlled-release fertilizers have been used extensively in recent years. Slow/controlled-release fertilizers can regulate the rate, timing and duration of nutrient release to meet crop demand based on different crops and soil types, so they could improve nutrient use efficiency and reduce environmental pollution [9]. As part of the recovery of both N and P from sewage or manure, struvite has gained much attention over the past decade [10]. As early as 1857, Murray proposed the use of struvite as a plant fertilizer due to the slow nutrient release and the low solubility in water [11]. Achat et al. (2014) [12] assessed the plant availability of P in struvite using isotopic labeling techniques, and found that it is feasible to substitute commercial fertilizers such as calcium phosphate primary with P recycled from pig manure and dairy effluents.

Struvite is crystalline compounds composed of magnesium ammonium phosphate ($MgNH_4PO_4 \cdot 6H_2O$), which is actually a mixture of two cations (magnesium, ammonium) and one anion (phosphate). Many studies have shown that struvite are more efficient in utilizing P. Kataki et al. [13] showed struvite can be slowly assimilated by the soil solution and nutrients release gradually, thus ensuring a steady supply of nutrients to plants and improving PUE. In addition, the release rate of P from struvite may more closely match the P uptake capacity of crop roots [14]. Organic acids have a strong effect on the dissolution rate of P from struvite, and plant species with dense root systems secrete large amounts of organic acids will be more efficient at absorbing P from struvite particles [15]. Predictably, struvite applied to paddy or wheat fields could benefit the crop P uptake and reduce the P loss due to the rice and wheat roots can secret large amounts of organic acid. However, a recent meta-analysis showed that over the past 57 year period since the first study in 1962, 70% of publications were greenhouse studies, whereas field studies only accounted for 8%, and most studies originated in Europe and Central Asia [15]. Little was known about the effect of struvite applied in rice–wheat rotation fields in China with different soil types under intensified cultivation mode.

Therefore, a two-year field experiment was conducted in a rice–wheat rotation system. The aims of the study were to: (1) determine the effect of struvite substituting traditional P fertilizer on crop growth, crop uptake and use efficiency of N and P; (2) study the effect of a P rate reduction on crop yield, P uptake and soil P availability; (3) explore a sustainable P management strategy to achieve the double win of food production and environment protection.

## 2. Materials and Methods

### 2.1. Experimental Site Description

A two-year field experiment was conducted from May 2020 to May 2022 at Fuzhuang site (32°06′ N, 119°06′ E) in Nanjing, Jiangsu Province, eastern China. This experimental field is a typical rice–wheat rotation system of rice in summer (June to October) and wheat in winter (November to May). The region is characterized by a typical subtropical climate with an annual average air temperature of 15.4 °C and precipitation of 1060 mm. The soil is classified as Horse liver soil in China, and as Gleyi-Stagnic Anthrosol according to the worldwide classification systems. The main properties of the soil are as follows: pH 6.22, organic matter content of 28.7 g/kg, available P content of 14.9 mg/kg and total nitrogen content of 1.94 g/kg.

### 2.2. Experiment Design and Field Management

Five treatments with three replicated plots (6 m × 6.7 m) were established in a completely randomized block design: farmer's P fertilization treatment (FP), struvite substitution of 100% P fertilizer (SP), struvite substitution with 50% P rate reduction (RSP), no application of N fertilizer (N0), no application of P fertilizer (P0). Details of the applied chemical fertilizers are presented in Table 1. The struvite used in this experiment was recycled from chicken manure produced by Shanghai Phosmag Environmental Technology Co., Ltd. (Shanghai, China), and its composition was shown in Table 2 and the purity was 94.4%. The heavy metal type and measure method is according to the Chinese national standard GB38400-2019. N content was determined by post-distillation titration, P content was determined by plasma emission spectrometry, K content was determined by flame photometry, Ca, Mg, Pb, Cd and Cr were determined by inductively coupled plasma emission spectrometry, Hg and As were determined via atomic fluorescence spectrometry.

**Table 1.** Two-year application of fertilizer N and P during the rice and wheat seasons.

| Treatment | N (kg/ha) (Rice/Wheat Seasons) | $P_2O_5$ (kg/ha) | |
|---|---|---|---|
| | | Rice Season | Wheat Season |
| FP | 240 | 70 | 96 |
| SP | 240 | 70 | 96 |
| RSP | 240 | 35 | 48 |
| P0 | 240 | 0 | 0 |
| N0 | 0 | 70 | 96 |

**Table 2.** Component and heavy metal analysis of struvite used in the experiment.

| | Component | Test Results |
|---|---|---|
| Composition (%) | $P_2O_5$ | 26.4 |
| | N | 5.0 |
| | Mg | 12.0 |
| | Ca | 0.065 |
| | $K_2O$ | 0.21 |
| Heavy metals (mg/kg) | As | undetected |
| | Hg | undetected |
| | Pb | 0.12 |
| | Cd | undetected |
| | Cr | 1.7 |

The N rate is same for all treatment except for N0 treatment and set as the recommended rate of 240 kg N/ha for both rice and wheat. The Urea was used as N fertilizer. The P rate of FP, SP and N0 was the same with the recommended rate of 70 kg $P_2O_5$/ha for rice and 96 kg $P_2O_5$/ha for wheat. The common P fertilizer of calcium superphosphate was used in FP and N0 treatments, but struvite was used as P fertilizer in SP and RSP treatments. When application, the N contained in the struvite was also calculated into

the total N rate. N fertilizer was split three times in the rice season (30% at transplanting, 30% at the tillering stage and 40% at the panicle initial stage) and four times in wheat season (40% at sowing, 20% at overwintering, 20% at regrowth and 20% at spike stage). All the P fertilizer was applied at transplanting or sowing. Rice seedlings of Nangeng 3908, a local Japonia variety, were transplanted at the middle of June, and harvested in the end of October. After harvesting, winter wheat seed (variety Yangmai 28) was directly sown to the soil surface, and harvested at the end of May. All aspects of field management, including the field preparation, irrigation, diseases and weed control, followed the local cultivation practices.

### 2.3. Sampling and Measurements

### 2.3.1. Crop Measurements

Before rice and wheat harvest, representative crop samples were taken from each plot to investigate the number of effective spikes, and then the plots were harvested individually for yield measurements. Destructive sampling was also carried out and the retrieved plant samples were oven-dried at 70 °C to a constant mass, weighed to determine aboveground dry biomass, and ground into powder (<0.149 mm fragments) for total N and total P analyses. N content in plant samples was analyzed by the micro-Keldjahl method, and P was determined using a UV-2600 spectrophotometer (Shimadzu Corporation, Kyoto, Japan) after strong-acid digestion. N and P use efficiency (NUE and PUE) was calculated according to Cassman et al. (1998) [16]. In order to identify the nutrient surplus status, apparent N or P balance was estimated using the deficit of the input and output of N or P in the whole system. The detail equations are as follows:

$$NUE = (N\ uptake\ X - N\ uptake\ 0)/N\ application \times 100\% \tag{1}$$

$$PUE = (P\ uptake\ X - P\ uptake\ 0)/P\ application \times 100\% \tag{2}$$

$$N\ apparent\ balance = fertilizer\ input - plant\ N\ uptake \tag{3}$$

$$P\ apparent\ balance = fertilizer\ input - plant\ P\ uptake \tag{4}$$

where NUE or PUE is the N or P use efficiency, N or P uptake X (kg/ha) is the N or P uptake by crop in fertilized area, and N or P uptake 0 (kg/ha) is the N or P uptake by treatment without N or P application.

### 2.3.2. Soil Analysis

After each crop harvest, soil samples were taken immediately from the surface layer (0–20 cm). For each plot, six soil cores were collected following the "S" pattern and sieved (2 mm) to obtain a composite sample after removing stones and plant and animal residues. The samples were air-dried for analyzing total N, soil available P, organic matter, soil available K and exchangeable calcium and magnesium. Soil available P was determined using a UV-2600 spectrophotometer (Shimadzu Corporation, Kyoto, Japan), after extracting with 0.5 mol/L NaHCO$_3$. Total N was analyzed using an vario MACRO cube elemental analyzer (Elementar Corporation, Frankfurt, Germany), and soil organic matter was determined via the potassium dichromate method. Available potassium was determined using BWB flame photometers (BWB Technologies, Newbury, UK) after extracting with ammonium acetate. An inductively coupled plasma spectrometer (ICP-OES) (Perkin Elmer, Waltham, MA, USA)was used as exchangeable calcium and magnesium detection means, and ammonium acetate was used as ion-exchange extractant.

### 2.4. Data Analysis

Data statistical analyses and graphing were operated by SPSS v27.0 (IBM Co., Armonk, NY, USA) and OriginPro 2019 (OriginLab, Northampton, MA, USA). One-way analyses of variance (ANOVA) were used to assess the statistically significant difference in crop

performance and soil properties among different treatments based on the Duncan's multiple range test ($p < 0.05$). There were 15 groups per season and 60 groups all seasons, fertilizer application was a single variable between each treatment. Error calculations were calculated via Excel 2019 (Microsoft, Downtown Bellevue, WA, USA).

## 3. Results and Discussion

### 3.1. Crop Yield and Aboveground Biomass

Figure 1 shows the yield of rice and wheat under five different treatments during the observation period. The SP treatment had the highest yield among the five treatments, while N0 treatment had the lowest yield, significantly lower than other four treatments. No significant differences were observed in both rice and wheat yield among FP, SP and RSP treatments, although 50% less P rate was applied in RSP treatment. Struvite substitution has the potential to promote the yield, and the total yield of SP treatment in two crop years increased 5.1% compared with FP treatment. It should be noted that the rice yield did not decrease significantly, but the wheat yield in both years decreased significantly in P0 treatment when comparison with FP and SP treatment.

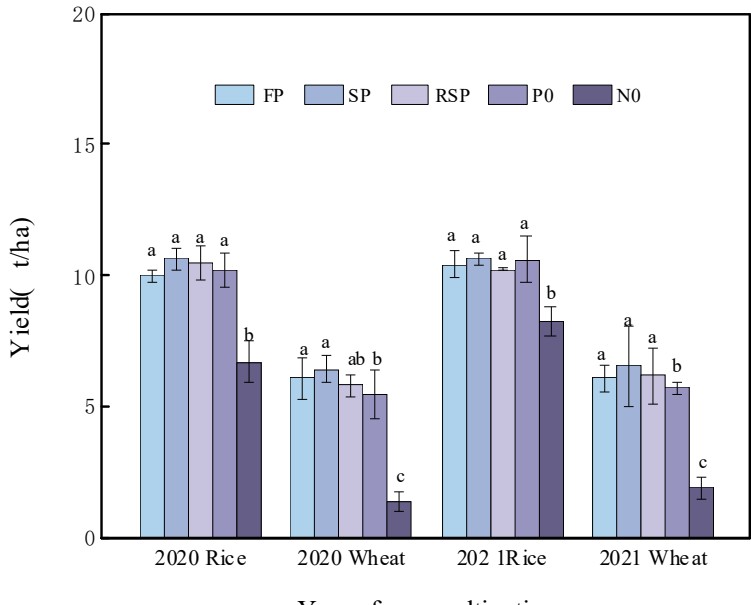

**Figure 1.** Grain yield under different treatments during the observation period. Note: Different lowercase letters in the graphs indicate significant differences between different treatments ($p < 0.05$), according to Duncan's multiple range test. FP: conventional fertilization; SP: struvite substitution of 100% P fertilizer; RSP: 50% P reduction with struvite substitution; N0: no application of N fertilizer; P0: no application of P fertilizer.

The same trends were found for the aboveground biomass; no significant differences were observed among FP, SP and RSP treatments in both rice and wheat season in both years; and SP treatment was the highest and N0 treatment was the lowest (Figure 2). For P0 treatment, the difference in aboveground biomass was also not significant when compared with FP, SP and RSP treatments, and significantly higher than N0 treatment in 2020 rice, 2021 rice and 2021 wheat.

### 3.2. Plant N and P Uptake and Fertilizer Use Efficiency

Struvite substitution significantly promoted the plant P uptake. Compared with FP treatment, the plant P uptake in SP treatment increased 7.6–46.4% and significantly higher than FP in 2020 wheat season (Table 3). The plant P uptake of RSP treatment with 50% P reduction did not show a decrease compared with FP; however, an increase of 10.7% in 2020 wheat, 25.9% in 2020 rice and 18.9% in 2021 wheat was observed although the difference

was not significant. No P application treatment could achieve 85.4–93% of plant P uptake in FP treatment, indicated that the soil P supply can meet the plant P need. Plant P uptake of N0 treatment significantly decreased compared with other treatments other than P0, although the P rate was the same with these treatments. PUE of FP treatment ranged from 3.3% to 16.3%, lower in the wheat season than that in the rice season (Figure 3). Struvite substitution significantly improved the PUE and the PUE of SP ranged from 12.3–26.9%, was 1.52 to 5.6 times of FP treatment. The PUE of the RSP treatment was the highest due to the 50% P reduction, and the value was as high as 49% in the 2020 rice season, 34.3% in 2020 wheat, 26.8% in 2021 rice and 30.0% in 2021 wheat, respectively.

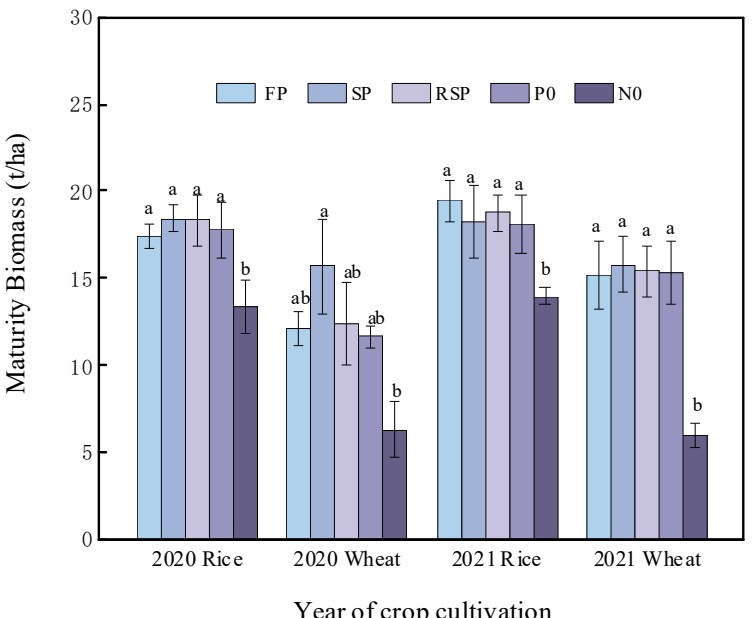

**Figure 2.** Biomass at maturity under different treatments during the observation period. Note: Different lowercase letters in the graphs indicate significant differences between different treatments ($p < 0.05$), according to Duncan's multiple range test. FP: conventional fertilization; SP: struvite substitution of 100% P fertilizer; RSP: 50% P reduction with struvite substitution; N0: no application of N fertilizer; P0: no application of P fertilizer.

Furthermore, we found that struvite substitution also improved the N uptake. The plant N uptake of SP increased 4.9–58% compared to FP treatment with the same N and P rate. RSP treatment also showed an increased tendency in comparison with FP treatment. No P application had little influence on N uptake, and the difference between P0 and FP is not significant. Therefore, the NUE of SP treatment was the highest, then followed by RSP and FP.

*3.3. Soil Nutrient Status*

Sustainable agriculture requires fertile soils with good chemical and physical properties that support root growth. Compared to FP treatment, the soil available P content significantly increased in struvite application treatments, especially in the first rice season, the soil available P content almost doubled in SP and increased by 68% in RSP with 50% less P input (Figure 4). In comparison with the initial value before experiment, the available P content was significantly decreased in the FP treatment but increased in the SP treatment with the same P input using struvite, while remained relatively constant in the RSP treatment. The soil available P content of P0 treatment was the lowest but there was no significant difference with FP treatment. There is no significant difference in the other soil nutrient indicators between the treatments (Table 4). Compared with the initial value before experiment, soil pH, organic matter, available K, Exchangeable Ca and Mg remained stable after two years' fertilization, but the soil TN showed a decreased tendency.

**Table 3.** Plant N and P content and uptake for rice and wheat of the different treatments in the two years.

| Crop Season | Treatment | Plant N Content (g/kg) | Plant P Content (g/kg) | Total N Uptake (kg/ha) | Total P Uptake (kg/ha) |
|---|---|---|---|---|---|
| 2020 rice | FP | 19.97 ab | 4.53 a | 177.1 (19.1) ab | 40.1 (4.2) ab |
| | SP | 22.12 a | 4.81 a | 206.3 (13.7) a | 45.1 (3.6) a |
| | RSP | 21.07 a | 4.77 a | 197.5 (42.0) a | 44.4 (7.6) a |
| | P0 | 19.88 ab | 4.40 a | 177.9 (1.9) ab | 36.9 (5.7) ab |
| | N0 | 16.18 b | 4.20 a | 91.3 (11.5) b | 27.8 (4.5) b |
| 2020 wheat | FP | 21.99 a | 3.39 a | 145.9 (10.6) ab | 20.1 (3.6) b |
| | SP | 25.84 a | 3.19 a | 230.7 (40.7) a | 29.4 (3.5) a |
| | RSP | 18.88 a | 3.64 a | 140.4 (18.8) ab | 25.3 (6.9) ab |
| | P0 | 25.35 a | 2.65 b | 143.5 (32.3) ab | 18.1 (2.1) b |
| | N0 | 21.94 a | 3.45 a | 65.4 (53.4) b | 21.1 (4.4) ab |
| 2021 rice | FP | 19.09 a | 3.46 a | 178.0 (19.8) a | 34.2 (2.2) a |
| | SP | 19.98 a | 3.78 a | 192.1 (20.7) a | 36.8 (2.2) a |
| | RSP | 18.89 a | 3.51 a | 188.2 (26.5) a | 33.3 (4.0) a |
| | P0 | 19.35 a | 3.19 a | 180.7 (27.8) a | 29.2 (4.1) ab |
| | N0 | 14.25 b | 3.39 a | 102.0 (11.6) b | 24.2 (1.9) b |
| 2021 wheat | FP | 21.12 a | 4.00 a | 164.3 (20.0) a | 25.9 (2.7) a |
| | SP | 21.02 a | 3.56 a | 172.3 (20.4) a | 29.7 (4.9) a |
| | RSP | 21.15 a | 4.31 a | 169.6 (11.3) a | 30.8 (1.8) a |
| | P0 | 19.88 a | 3.34 ab | 158.4 (13.2) a | 24.5 (3.0) a |
| | N0 | 21.73 a | 2.99 b | 67.1 (16.7) c | 12.5 (3.0) b |
| All seasons | FP | 20.54 a | 3.84 a | 166.3 (12.9) b | 30.1 (7.6) ab |
| | SP | 22.24 a | 3.84 a | 200.4 (21.3) a | 35.2 (6.4) a |
| | RSP | 20.00 a | 4.06 a | 173.9 (21.8) ab | 33.5 (6.9) a |
| | P0 | 21.12 a | 3.39 b | 165.2 (15.1) b | 27.2 (6.9) ab |
| | N0 | 18.52 b | 3.50 ab | 81.5 (15.7) c | 21.4 (5.6) b |

Note: Values in brackets are standard errors. Different lowercase letters in a row indicate significant difference across the five treatments in the same season at $p < 0.05$, according to Duncan's multiple range test. FP: conventional fertilization; SP: struvite substitution of 100% P fertilizer; RSP: 50% P reduction with struvite substitution; N0: no application of N fertilizer; P0: no application of P fertilizer.

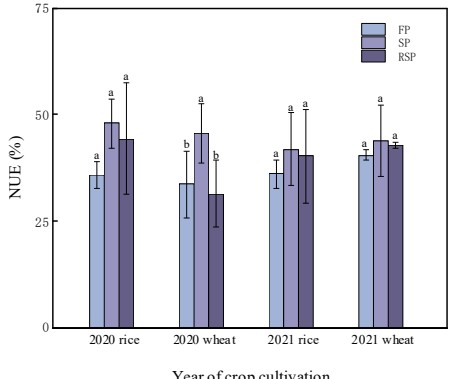
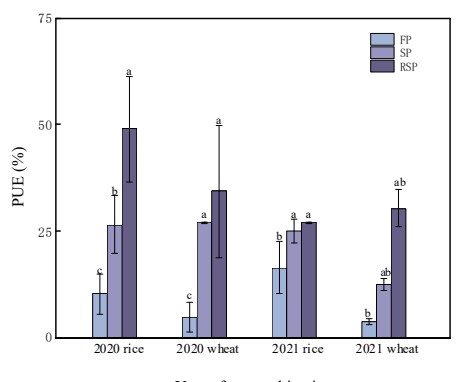

**Figure 3.** NUE (%) and PUE (%) under different treatments (except N0 and P0) during the observation period. Note: Different lowercase letters in the graphs indicate significant differences between different treatments ($p < 0.05$), according to Duncan's multiple range test. NUE and PUE means the N and P use efficiency. FP: conventional fertilization; SP: struvite substitution of 100% P fertilizer; RSP: 50% P reduction with struvite substitution; N0: no application of N fertilizer; P0: no application of P fertilizer.

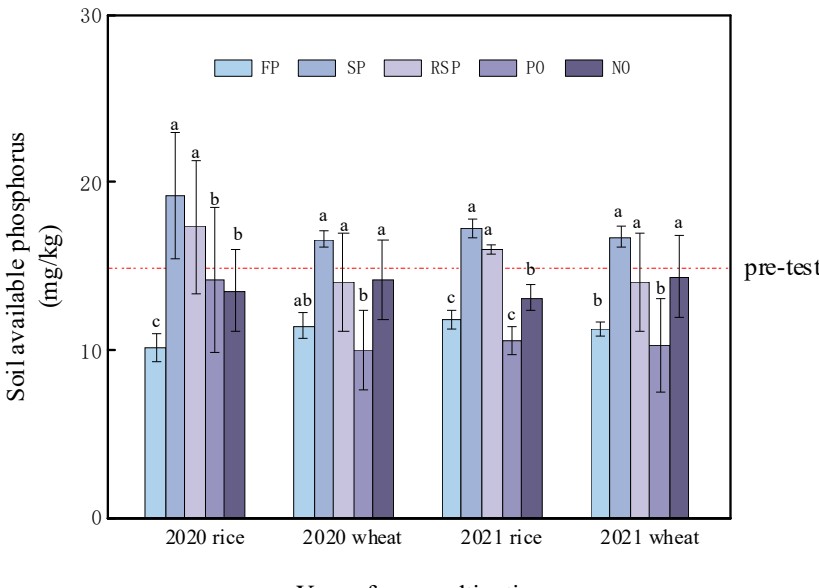

**Figure 4.** Soil available P content under different treatments during the observation period. Note: The red dashed lines represent soil available P before the experiment. Different lowercase letters in the graphs indicate significant differences between different treatments ($p < 0.05$), according to Duncan's multiple range test. FP: conventional fertilization; SP: struvite substitution of 100% P fertilizer; RSP: 50% P reduction with struvite substitution; N0: no application of N fertilizer; P0: no application of P fertilizer.

**Table 4.** Indicators of soil-related nutrients at maturity for rice and wheat of the different treatments in the two years.

|  |  | pH | Organic C (g/kg) | TN (g/kg) | Available K (mg/kg) | Exchangeable Ca (cmol/Kg) | Exchangeable Mg (cmol/Kg) |
|---|---|---|---|---|---|---|---|
| 2020 rice | FP | 6.27 (0.11) a | 23.53 (2.19) b | 1.57 (0.09) b | 70.27 (8.23) b | 17.98 (1.60) a | 3.79 (0.13) a |
|  | SP | 6.25 (0.14) a | 29.38 (1.43) a | 1.97 (0.03) a | 82.71 (11.98) ab | 17.04 (0.58) a | 3.76 (0.15) a |
|  | RSP | 6.17 (0.31) a | 30.89 (4.13) a | 2.05 (0.34) a | 69.03 (3.80) b | 17.25 (0.61) a | 3.69 (0.09) a |
|  | P0 | 6.03 (0.16) a | 31.87 (4.20) a | 2.02 (0.19) a | 76.99 (3.92) b | 17.64 (1.33) a | 3.50 (0.32) a |
|  | N0 | 6.38 (0.13) a | 28.02 (3.14) a | 1.91 (0.17) a | 99.13 (12.84) a | 17.41 (0.20) a | 3.65 (0.12) a |
| 2020 wheat | FP | 6.33 (0.14) a | 17.28 (1.20) b | 1.11 (0.07) a | 74.33 (5.31) a | 17.85 (1.21) a | 3.87 (0.21) a |
|  | SP | 6.28 (0.01) a | 20.47 (2.78) ab | 1.29 (0.14) a | 86.33 (6.65) a | 18.7 (0.19) a | 4.27 (0.34) a |
|  | RSP | 6.33 (0.06) a | 22.19 (3.56) a | 1.36 (0.19) a | 84.33 (8.26) a | 18.70 (2.17) a | 3.95 (0.27) a |
|  | P0 | 6.37 (0.18) a | 21.89 (2.90) a | 1.34 (0.13) a | 86.67 (2.05) a | 17.31 (1.07) a | 3.76 (0.17) a |
|  | N0 | 6.29 (0.03) a | 19.30 (1.49) ab | 1.28 (0.15) a | 88.67 (10.40) a | 18.89 (0.49) a | 3.93 (0.05) a |
| 2021 rice | FP | 6.17 (0.06) a | 21.24 (0.46) a | 1.57 (0.09) a | 77.94 (1.72) a | 17.86 (0.98) a | 3.36 (0.19) a |
|  | SP | 6.20 (0.02) a | 20.38 (0.71) a | 1.97 (0.03) a | 83.38 (2.30) a | 17.43 (0.43) a | 3.65 (0.09) a |
|  | RSP | 5.98 (0.16) a | 20.89 (1.07) a | 2.05 (0.34) a | 80.8 (4.45) a | 17.49 (0.85) a | 3.58 (0.15) a |
|  | P0 | 6.11 (0.11) a | 20.75 (0.80) a | 2.02 (0.19) a | 78.92 (2.68) a | 17.38 (0.29) a | 3.27 (0.13) a |
|  | N0 | 6.31 (0.10) a | 20.69 (0.25) a | 1.91 (0.17) a | 79.13 (1.34) a | 17.12 (0.24) a | 3.52 (0.19) a |

**Table 4.** *Cont.*

|  |  | pH | Organic C (g/kg) | TN (g/kg) | Available K (mg/kg) | Exchangeable Ca (cmol/Kg) | Exchangeable Mg (cmol/Kg) |
|---|---|---|---|---|---|---|---|
| 2021 wheat | FP | 6.22 (0.08) a | 18.95 (0.40) a | 1.31 (0.07) a | 78.67 (0.94) a | 18.41 (0.93) a | 3.78 (0.06) a |
|  | SP | 6.28 (0.01) a | 20.14 (1.15) a | 1.32 (0.10) a | 83.00 (5.35) a | 18.59 (0.07) a | 3.97 (0.04) a |
|  | RSP | 6.32 (0.06) a | 21.19 (1.21) a | 1.26 (0.05) a | 81.00 (3.56) a | 18.27 (1.06) a | 4.01 (0.15) a |
|  | P0 | 6.24 (0.06) a | 19.89 (0.77) a | 1.34 (0.06) a | 86.67 (2.05) a | 17.59 (0.72) a | 3.80 (0.06) a |
|  | N0 | 6.30 (0.03) a | 19.63 (1.11) a | 1.26 (0.05) a | 82.00 (5.72) a | 18.91 (0.42) a | 3.92 (0.05) a |

Note: Values in brackets are standard errors, followed different lowercase letters in the table indicate significant difference across the five treatments in the same season at $p < 0.05$, according to Duncan's multiple range test. FP: conventional fertilization; SP: struvite substitution of 100% P fertilizer; RSP: 50% P reduction with struvite substitution; N0: no application of N fertilizer; P0: no application of P fertilizer.

### 3.4. N and P Apparent Balance

The apparent N balance showed that there was little difference between conventional fertilization and struvite application (Figure 5). Struvite substituting conventional P fertilizer reduced the N balance through increased the N uptake, which indicated that the N loss was reduced. Except for the P0 treatment, all treatments showed a positive P balance, with a P surplus of 53.8 kg/ha in the FP treatment, suggesting that the P rate in the FP treatment has exceeded the needs of crops. Thus, the amount of P should be reduced to prevent environmental risks. The P balance of SP treatment was slightly lower than that of FP treatment. In contrast, the P balance of the RSP treatment was 6.6 kg/ha, which was close to zero, significantly lower than that of FP and SP treatments.

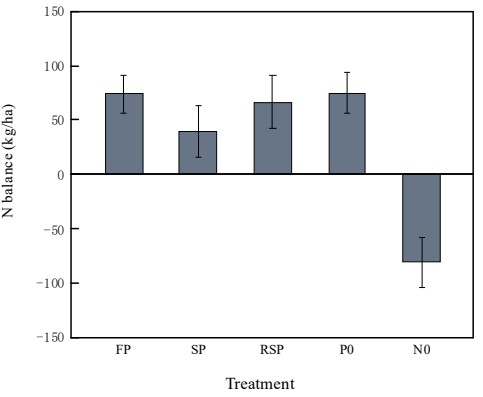 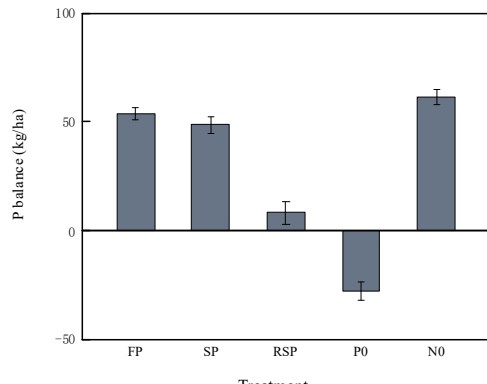

**Figure 5.** N and P apparent balance under different treatments during the observation period. FP: conventional fertilization; SP: struvite substitution of 100% P fertilizer; RSP: 50% P reduction with struvite substitution; N0: no application of N fertilizer; P0: no application of P fertilizer.

## 4. Discussion

There have been many reports evaluating the effect of struvite as a fertilizer on variety of crops, soil types and climate. Kataki et al. (2016) [13] summarized the effect of struvite on 20 plant varieties and found no significant differences between P in struvite and P in other phosphate fertilizers, and 14 out of the 19 studies reported superior or comparable effects of struvite over the chemical fertilizer on crop growth. In this study, a similar aboveground biomass and grain yield was observed in struvite treatments in rice and wheat for two successive years in a field study in China (Figures 1 and 2); meanwhile, the crop P uptake and N uptake were both increased by 7.6–46.4% and 4.9–58% compared with the conventional P fertilizer treatment with calcium superphosphate at the same N and P rate (Table 3). This is in agreement with the previous study, which reported increased total P uptake in wheat using struvite recovered from dairy wastewater in pot experiment [17], and in corn and soybean in a field experiment [18]. Additionally, it was

shown that aboveground P concentrations and P uptake response ratios were significantly affected by soil pH and decreased with increasing pH [15], but in this study, the soil pH of the treatments was almost unchanged from the start of the experiment (Table 4).

The higher P uptake due to struvite application was mainly attributed to most of the P in struvite being acid soluble [11]. Rech et al. (2019) [19] reported that the citrate solubility of struvite produced from poultry manure, swine manure and municipal wastewater was 18%, 28% and 29%, respectively, whereas water solubility was lower and more constrained at 2–3%. Citrate solubility means struvite dissolution could be driven by organic acids exuded by plant roots, especially for the grain crop of rice and wheat with dense roots [14]. For traditional P fertilizer with high solubility, phosphate ions that are not absorbed by crops are quickly fixed by the soil through the combination with such as iron and calcium to form insoluble phosphates, so the P availability was decreased. However, the solubility of struvite in water is low, the P in struvite was gradually released with the crop growth through the organic acid exudated by crop roots, and the P supply was synchronized with the P demand [20]. This also ensure the high P use efficiency. Therefore, the P concentration is higher in plants grown with struvite than in plants grown with other P fertilizers, not only in our study but also in other studies [21,22]. In our study, we can see that PUE was significantly increased from no more than 10% in traditional P fertilizer treatment to 22.7% in SP treatment on average (Figure 3). In addition, the Mg present in struvite may be another contributor of the higher P uptake due to the synergistic effect of Mg on P absorption [13]. This is verified by Rasul et al. (2011) [23], who observed Mg application could improve the PUE of wheat by over 55%. So, struvite application is advantageous in soils and crops with high Mg and P demand, and under acid and neutral soil conditions.

Improved N uptake and NUE were also observed in our study in both rice and wheat with struvite application treatment (Table 3, Figure 3). This mainly related with the slow-released N in struvite due to its low water solubility. In our study, additional N was applied using urea to ensure the total N rate was optimal for rice and wheat growth. The N brought by struvite was about 6.2% and 8% of the total N input in rice and wheat season, respectively. This means that the treatments with struvite were the combination of slow-released N fertilizer and fast-released N fertilizer. In accordance, the N supply was optimized compared to FP treatment with only fast-released fertilizer urea and plant N uptake was increased. Lower plant N uptake with the application of struvite without additional N but greater N uptake was observed when using struvite with additional N for maize and annual ryegrass [24]. This also indicate that struvite must be combined with additional N fertilizer due to its low N content and low N dissolution rate to ensure the crop yield. Furthermore, the potential of struvite to lower N loss risk may be another reason for the greater N uptake. Rahman et al. (2011) [25] observed that N leaching loss was only 2.0% of the total N supplied in struvite-treated soil, while it was as high as 7.1% of the total N applied in common chemical fertilizer (fused super phosphate+ urea)-treated soil. Ahmed et al. (2016) [26] identified less N leaching from struvite compared with conventional fertilizers such as superphosphates across 10 studies.

The soil nutrient apparent balance reflects the surplus or deficit status of soil nutrient in the current season, a positive value indicates the over input of nutrients and the nutrient will be lost to the environment or some will be saved in the soil, while a negative value indicated the insufficient input and the soil nutrient will be exhausted [27]. In an ideal state, the optimal fertilizer input should be equal to the crop uptake or the amount of crop removal from the fields. The utilization of P by crops is relatively low due to the strong adsorption, rapid precipitation and fixation into unusable forms of P. A global analysis of PUE in cereal crops showed grain and aboveground PUE were on average 9.1% and 12.4%, respectively, which indicated the great potential to improve PUE by minimizing P fertilization [28]. Numerous studies showed that there is a positive relationship between soil test P levels and P loss to water [29]. Considering the limited P resources, reducing P application was thought to be the first option to maintain a low P accumulation in soil and reduce the P loss risk. A multi-year study of rice–wheat rotation in Taihu region in

China by Wang (2016) found that no P application in the rice season had no effect on the yields of rice and wheat over 4 years, but it increased the PUE notably compared with P fertilization in both rice and wheat seasons; however, the P reduction in the wheat season may lower the yield [30]. In our study, the P reduction by half with struvite did not affect the crop growth and grain yield, but it slightly improved the P uptake and soil available P content compared to FP treatment (Figures 1 and 4, Table 3).

As struvite persists in the soil for several months after application, it has been proven that struvite does significantly increase soil P levels above optimum levels as a result of the inherent low solubility of struvite [31]. In subsequent crop growing seasons (i.e., the year after application), the remaining struvite will continue to provide P for crops. This is the reason why even if the P input from struvite is reduced, the available P content of the soil did not decrease compared to the initial value in our study. However, the apparent P balance of FP treatment was the highest, but the soil P availability was the lowest and even lower than the initial value at the beginning of experiment, which indicated that more P was lost and the residual P was turned into insoluble P. Gong et al. (2022) [32] found that mineral P fertilizer types have an effect on the PUE and P loss, and diammonium phosphate and monoammonium phosphate were more beneficial than the single superphosphate, triple superphosphate and calcium magnesium phosphate based on a meta-analysis. Meanwhile, struvite was found be advantageous for better P uptake than diammonium phosphate [26]. Apart from the fertilizer type, managing soil fertility around critical P levels may maximize PUE [33]. In our study, the P apparent balance of RSP treatment was near zero and obviously lower than SP and FP (Figure 5), which means the soil P supply was nearly equal to the crop P demand and soil P pool was not depleted. Accordingly, the PUE of the RSP treatment was much higher than that of the other treatments (Figure 3). Thus, the P rate reduction combined with struvite substitution seems to be a better sustainable option for improving PUE and maintaining the crop productivity and optimal soil P availability compared with traditional P fertilizer.

Struvite is an emerging and recyclable low water-soluble P fertilizer, and it has already shown a great potential in improving crop N and P uptake and crop growth or yield, and in reducing the nutrient loss risk. The strategy of struvite as a fertilizer would have a profound impact on global food security and environmental protection. In future research, we should focus on how to promote the struvite in practical applications. The cost–benefit analysis should be performed considering the environmental benefit in view of the associated benefits of the struvite recovery process (cost savings from sludge volume reduction and pre-requisite for chemical treatment; conservation of limited P resources and the safe disposal of waste) due to the expensive production of struvite compared with the extraction of phosphate rock fertilizer [34]. Furthermore, the adverse effect of struvite application should be comprehensively evaluated before extension due to a potential increase in soil pH [14] and as its recycling wastewater usually contains certain heavy metals (Cd, Cu, Cr, Ni, Pb, Zn, etc.) and organic pollutants. However, in our study, the soil pH of struvite application almost remained unchanged compared with FP and the initial value in a neutral soil (Supplementary Table S1), and the heavy metal content in struvite derived from chicken manure sludge was lower than the established criteria of agricultural inputs in China, and various studies had also already confirmed the safety of struvite [34–37]. Last, the optimal application of struvite requires additional research based on crop species, cropping rotations and soil status to ensure the crop productivity and lower the loss risk of N and P. More research studies in this field over a longer term are still needed, in particular for economically important grain crops.

## 5. Conclusions

The overuse of P fertilizer was observed in the conventional nutrient management in rice–wheat rotation system in the lower region of Yangtze river in China. Struvite substitute conventional P fertilizer could increase the rice and wheat yield and both the use efficiency of N and P, while enhancing the soil available P content under the same N

and P application rate. A 50% P reduction using struvite as a P fertilizer has no significant effect on crop P uptake and final yield, but it improved the NUE and PUE and maintained a relative constant soil available P level with the P balance close to zero. Considering the sustainability of crop yield and soil fertility, resource use efficiency and environmental loss risk, it is suggested that the optimal P management strategy under the rice–wheat rotation system is to use struvite as a P fertilizer while reducing the P rate appropriately.

**Supplementary Materials:** The following supporting information can be downloaded at: https://www.mdpi.com/article/10.3390/agronomy13122948/s1. Supplementary Figure S1: Precipitation and average temperature in different months of 2020 to 2022 during the observation period. Supplementary Table S1: Precipitation and average temperature in different months of 2020 to 2022 during the observation period.

**Author Contributions:** Methodology, L.X. (Lihong Xue), L.Y. and S.L.; Formal analysis, J.W.; Investigation, J.W., T.H., Lixiang Xue and T.S.; Resources, L.X. (Lihong Xue); Data curation, J.W.; Writing—original draft, J.W.; Writing—review and editing, J.W., L.X. (Lihong Xue), T.H. and X.Z.; Visualization, J.W.; Supervision, L.X. (Lihong Xue), P.H. and L.Y.; Project administration, L.X. (Lihong Xue) and P.H.; Funding acquisition, L.X. (Lihong Xue) and S.L. All authors have read and agreed to the published version of the manuscript.

**Funding:** This work was financially supported by the National Key Research and Development Program of China (No. 2021YFD1700803), National Science Foundation of China (41877087) and the Province Key Research and Development Program of Jiangsu, China (D21YFD17008).

**Data Availability Statement:** Data are contained within the article.

**Acknowledgments:** The authors wish to acknowledge Shanghai Phosmag Environmental Technology Co., Ltd. (Shanghai, China) for providing the struvite and fund to carry out the experiment.

**Conflicts of Interest:** Author Sergey Lobanov was employed by the company Shanghai Phosmag Environmental Technology Co. Ltd. (Shanghai, China). The remaining authors declare that the research was conducted in the absence of any commercial or financial relationships that could be construed as a potential conflict of interest.

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
