# Peer review of "Struvite as P Fertilizer on Yield, Nutrient Uptake and Soil Nutrient Status in the Rice–Wheat Rotation System: A Two-Year Field Observation"

_agronomy, doi:10.3390/agronomy13122948_

Round 1
Reviewer 1 Report
Comments and Suggestions for Authors
Dear Editor and authors, the article has potential but needs several changes to be published. Furthermore, due to the low number of repetitions (3 blocks), there is the possibility of a serious error in the statistical part of the work. Some comments are provided below.
- In the introduction section, some paragraphs are long, please make the information more succinct, this is evident in paragraph 3
- In the introduction, a paragraph must be inserted talking about struvite, its chemical composition and from there why struvite can be more efficient in using P, there are already several works that do this.
- In the material and methods section, a figure must be provided with data on temperature and monthly rainfall during the entire experimental period and not just the average described in the text.
- In relation to the experimental design, due to the 5 treatments tested with 3 blocks, the degree of freedom of the residue is only 8, with the minimum acceptable being 12, just as the authors controlled the possibility of committing a type I error in their analysis of data and how can your inferences be validated by this low degree of freedom of the residue, which is below adequate?
- How do the authors control the effect that struvite can give due to the Mg contained in it? – the authors need to provide data on Mg absorbed by the plant and correlate whether the effect is due to higher Mg levels or not, since Mg is involved in several key processes, such as the activation of the ATP molecule.
- In the results section, the presentation of figures 1 and 2 is poor, the authors cannot place wheat and rice in the same figure, as they are different species, obviously the scales on the y axis are different, I suggest dividing the figure into 4 parts, separating by year and each species.
- The legends of the figures and tables need to be self-explanatory, and lack a lot of information, such as the insertion of letters for the comparison of the means test and what they mean.
- The discussion is long and tiring, in addition the paragraphs need to be restructured, they are very long
- The discussion section should not be subdivided into topics, it should be continuous
- The discussion section needs to establish a cause and effect relationship more clearly, focusing on the main properties of struvite and how it can increase the efficiency of P use, it is too generalized the article's discussion.
- The future challenge section and everything that is discussed must be provided in the last paragraph clearly and succinctly, using half of the text, because this paragraph is only to close the discussion.
Reviewer 2 Report
Comments and Suggestions for Authors
The present work evaluated the effects of struvite recycled from chicken manure as P fertilizer in a field two years rice-wheat rotation system. A total of five treatments were compared, including conventional fertilization, struvite substitution of 100% P fertilizer, 50% P reduction with struvite substitution, no application of N fertilizer and no application of P fertilizer. Grain yield, crop N and P uptake, N and P use efficiency and soil nutrient status were assessed. According to the results of this study, the substitution of conventional phosphorus fertilizer with struvite provides benefits to the crop even by reducing the P dose by half. This reduction in dosage improves the P balance and reduces the accumulation of P in the soil, with a consequent environmental benefit.
The work is very interesting, both for its results and for its nature as a field trial. Field studies, as the authors comment in the text, are scarce and always provide very valuable information.
From my side it would be published with a minor revision.
My specific comments are:
Line 31: in rice—wheat: one extra hyphen.
Line 109: There is an error in the units of measurement of soil available P. The correct for is mg/kg not g/kg.
Line 177-178: You said “the wheat yield in both years decreased significantly in P0 treatment when compare with FP treatment”. Also decrease significantly when compare with SP treatment.
Line 197: “…25.9% in 2020 rice and 18.9% in 2021 wheat…”.
Line 201: You write that plant P uptake of N0 decrease compared with FP, but decrease also compared with both struvite treatments (with the same dose or the 50%). Check it please.
Figure 3: Check letters of the graphs. They do not correspond to what is expressed in the text for P0 treatment.
Lines 232-237: This is a good introduction and explanation of nutrient balance, but you should consider including it in the discussion section as well as suggestions and results explanation in this paragraph.
Figure 4. Letters in the graphs are missing. Check it please.
Line 258: Figure.1.
Line 261: I think Table 2 is wrong. You probably refer to Table 3.
Line 277-278: The sentence is not well understood, it seems incomplete. Please review, please.
Line 315-316: You write: “slightly improve the P uptake and soil available P content compared to FP treatment (Figure 1, Table 3, Figure 4)”. I think the correct reference is (Table 3, Figure 3). Review it, please.
Lines 351-354: The sentences are not well understood, in my opinion you have to put them together in one sentence.
Line 360: “optical”: I think you meant to say "optimal".
In Materials and Methods section you must to describe, briefly, the determination method of N and P content of plants. In addition, although the results can be assumed with the results of N and P uptake, you should express the N and P content of the plant material in the results section, either in a table in that section or put it as supplementary material.
Reviewer 3 Report
Comments and Suggestions for Authors
Comments and suggestion for Authors in attached file.

Reviewer 4 Report
Comments and Suggestions for Authors
The authors investigate the impacts of adding struvite as an alternative P fertilizer and measure the impact on a rice-wheat rotation over two years. They have a P and N control in each treatment and measure the yield and P and N use efficiency in all treatments. The work is applicable to the field and has some novel insights. Overall the science is good; however, some edits should be completed before accepting the manuscript.
The last paragraph of the introduction should include more information on the experiment done in this manuscript.
What were the average temperature and rainfalls for the two years of this experiment? This would be useful to have because this is a field trial.
What is Horse liver soil? Please add the World Reference Base for Soil Resources classification.
What technique was used to determine the elemental concentrations in Table 2?
In Figure 1 please adjust the letters indicating statistical significance in the N0 treatment so they are in the correct section of the four bars.
For Figures 1 and 2, it is hard to compare the height of the bars this way. Consider stacking side by side instead of vertically like this. Please also include the meaning of any abbreviations used in the figure descriptions.
pH is important in the P availability. It should be mentioned in the main text and not only in the supplemental. Because there is only 1 table in the supplemental, consider improving the readability of this table and including it in the main text. You could include each error in () instead of +/- with a note in the description. This will make reading the table much easier.
Please consider using fewer significant digits when reporting percentages.
Please consider splitting some of the larger paragraphs based on the main ideas to improve the readability of the text.
Comments on the Quality of English Language
The manuscript would benefit from some editing for grammar and organization.
